# On the Fence: The Impact of Education on Support for Electric Fencing to Prevent Conflict between Humans and Baboons in Kommetjie, South Africa

**DOI:** 10.3390/ani13132125

**Published:** 2023-06-27

**Authors:** Debbie Walsh, M. Justin O’Riain, Nicoli Nattrass, David Gaynor

**Affiliations:** 1Institute for Communities and Wildlife in Africa (iCWild), University of Cape Town, Private Bag, Rondebosch, Cape Town 7701, South Africa; justin.oriain@uct.ac.za (M.J.O.); nicoli.nattrass@uct.ac.za (N.N.); 2Mammal Research Institute, University of Pretoria, Private Bag X20, Hatfield, Pretoria 0028, South Africa; dgaynor@iafrica.com

**Keywords:** baboon-proof fence, conservation management, education, evidence-based intervention, human–wildlife conflict, spatial overlap, stakeholders, urban spaces

## Abstract

**Simple Summary:**

Conservation often requires that people change their behaviour or accept interventions that seek to improve the conservation and welfare of wildlife and the safety and well-being of people. In this study, we show that a short educational video can improve community support for an intervention (electric fencing) that is advocated by experts as a sustainable and cost-effective intervention to keep baboons out of urban residential areas. By varying the timing of the video within a survey exploring support for an electric fence, we demonstrate the value of education, which increased the average marginal probability of support by 15 percentage points. Women were more likely to change their attitude to fences once apprised of the relevant facts than men. This study contributes to the emerging literature on the importance of education in managing conservation conflicts and the need for evidence-based interventions.

**Abstract:**

Few studies test whether education can help increase support for wildlife management interventions. This mixed methods study sought to test the importance of educating a community on the use of a baboon-proof electric fence to mitigate negative interactions between humans and Chacma baboons (*Papio ursinus*) in a residential suburb of the City of Cape Town, South Africa. An educational video on the welfare, conservation and lifestyle benefits of a baboon-proof electric fence was included in a short online survey. The positioning of the video within the survey was randomised either to fall before or after questions probing the level of support for an electric fence. The results showed that watching the video before most survey questions increased the average marginal probability of supporting an electric fence by 15 percentage points. The study also explored whether the educational video could change people’s minds. Those who saw the video towards the end of the survey were questioned again about the electric fence. Many changed their minds after watching the video, with support for the fence increasing from 36% to 50%. Of these respondents, the results show that being female raised the average marginal probability of someone changing their mind in favour of supporting the fence by 19%. Qualitative analysis revealed that support for or against the fence was multi-layered and that costs and concern for baboons were not the only relevant factors influencing people’s choices. Conservation often needs to change people’s behaviours. We need to know what interventions are effective. We show in the real world that an educational video can be effective and can moderately change people’s opinions and that women are more likely to change their position in light of the facts than men. This study contributes to the emerging literature on the importance of education in managing conservation conflicts and the need for evidence-based interventions.

## 1. Introduction

As human population size increases, competition for space and resources between people and wildlife is increasing, ultimately leading to conflicts amongst people on how best to manage wildlife populations [1,2,3]. Negative human–wildlife interactions have adverse impacts on individual animals, populations and biodiversity more generally, in addition to the health and livelihoods of affected human communities [4,5,6]. Negative interactions with wildlife were previously considered to be largely a “rural or agricultural” issue, but today, conflicts in peri-urban spaces are commonplace and include a range of animal taxonomic groups and contexts across the globe [7,8].

Baboons are considered one of the most difficult animals for humans to coexist with [9] and have high levels of negative interactions with humans throughout their distribution in both rural and urban areas [10]. The genus *Papio* is dextrous, agile, intelligent, social, has high dietary flexibility and wide habitat tolerance and is equally adept at running as it is climbing [11]. These attributes have allowed them to exploit human foods from a variety of sources, including crops, houses and even vehicles. Baboon ‘raiding’ behaviour can lead to severe losses for local rural economies, where it is estimated that in a single raiding event, baboons around Kibale National Park, Uganda, damaged up to 2774 m^2^ of crops [12]. A different study in Uganda examined the impact of crop-raiding by olive baboons (*Papio hamadryas anubis*) and found they were responsible for 70% of all crop damage events [13]. The most common approach to mitigate these negative impacts was guarding and chasing the baboons away, a costly approach as it requires constant vigilance.

Although wild primates are threatened due to habitat loss and fragmentation, access to high-quality anthropogenic food and the elimination of natural predators can have a positive effect on primate numbers [14,15]. Baboons are able to thrive in human-modified environments, accessing high-quality human foods associated with agriculture, tourism and human settlement, creating a negative relationship with humans [8,16,17]. This relationship, where baboons become habituated to a consistent supply of high-quality food, leads to an increase in their numbers and, thus, to further negative interactions with humans [9]. Increased levels of negative interactions can lead to increased mortality and injury [18], with mortality in baboons and vervet monkeys known to be the highest in troops that have frequent contact with humans [8,19,20]. Most species of baboon within the genus *Papio* are not listed as threatened or endangered [21], but this could change as human populations continue to expand.

Many management measures have been used to try and reduce baboon presence in urban and rural landscapes of Cape Town, including herding [22], light prisms [23], provisioning [17], virtual fencing [24], translocation [20] and the euthanising of habitual raiders that pose a risk to baboon and human health and well-being [25]. The use of baboon monitors, people employed to usher baboons away from residential areas in Cape Town, has been in use since 2001 to the present day and can reduce the time baboons spend in urban areas by up to 67% [22]. The research was carried out to test the efficacy of different deterrents aimed at reducing the overlap between human environments and baboon home ranges [26]. Reflective light prisms did not change the range used by the baboons [23] while bear bangers (thumb-sized cartridges that are propelled by pen-sized launchers into the air and make a loud bang for a short duration) were highly effective in deterring the troop from entering the residential area [26].

Electric fencing is a particularly effective but expensive method for minimising human–wildlife conflict [22]. It is a barrier that works by making a fence unclimbable by carrying a high voltage (5000–10,000 v) but low amperage (~0.120 amps) for 3 ms pulsed at 1 s intervals. This results in a disruption to neural and muscle transmission, meaning that baboons immediately lose their ability to grip and cannot climb over the fence. This aversive measure results in baboons learning to avoid climbing the fence because of the unpleasant shock experience. The fence acts as a direct obstacle to an animal’s movement and has been applied as a mitigation measure for many species [6] including bears, elephants and large carnivores [27].

Electric fencing has been used as a physical barrier to successfully restrict the movement of baboons, both within conservation areas and sanctuaries/zoos and out of designated exclusion areas such as farmlands [28]. The diverse mode of locomotion and agility of baboons enables them to traverse many man-made barriers, including ineffective electric fences. Properly designed electric fences can be extremely effective. An example is the Zwaanswyk baboon-proof fence around the residential suburb of Zwaanswyk in Cape Town [26]. The suburb consists of upmarket homes on large plots, many of which have fruit trees and vegetable patches, which are attractants for baboons. The Zwaanswyk fence was erected on the boundary between the suburb and Table Mountain National Park and is approximately 2.3 km in length and includes 12 electrified strands on the exterior surface of 10 × 10 cm Bonnox^®^ (Centurion, South Africa) mesh fencing, which acts as a physical barrier (Figure 1). The top part of the fence includes electrified wires that are angled at a 45° slope to form an anti-climb overhang. The fence is 2.4 m tall and extends below ground to prevent baboons and other wildlife (e.g., porcupines) from digging under the fence.

Baboon troop movements were recorded by a GPS collar before and after the installation of the fence to monitor how this intervention would affect baboon movement in relation to the suburb. Using kernel density estimates, it was shown that once fencing was erected around Zwaanswyk, the troop’s core use area did not overlap with the suburb at all [26]. An additional benefit of the fence is that no buffer zone was required, allowing the troop to forage right up to the fence and have maximal access to their natural land within their home range.

Despite the success of the Zwaanswyk fence, other suburbs on the Cape Peninsula experiencing high levels of negative interactions with baboons have not pursued baboon-proof fencing to deter baboons. This is despite conservation authorities investing considerable effort in education and awareness campaigns that are designed to inform and change people’s attitudes to conservation and conservation management strategies, including baboon-proof fencing. There is little empirical evidence in the peer-reviewed literature of how effective these initiatives are [29]. Brief educational interventions have, however, often been shown to be effective in clinical trial research. In one study, a randomised controlled trial design was used to test whether a brief multimedia psychoeducational intervention would affect the attitudes and interests of patients with cancer regarding clinical trial participation, where negative attitudes of patients are generally an important contributor to low participation rates [30]. It was found that patients who received the educational intervention, a 10 min DVD addressing misperceptions and concerns about clinical trials, subsequently showed a more positive attitude toward clinical trials and a greater willingness to participate in them. This inspired the initiative discussed in this paper, namely, the use of an educational video within a short online survey on attitudes towards fences and baboons in a community beset with negative interactions between humans and baboons.

Our goal was to test whether education influences attitudes towards an intervention that is recommended by independent experts in human-baboon interactions and the local conservation authorities.

## 2. Materials and Methods

### 2.1. Survey Design

This study employed mixed methods to understand attitudes towards electric fences as a solution to reducing spatial overlap between people and baboons and to explore whether an educational video could increase support. Attitudes were analysed quantitatively and qualitatively. The main quantitative research question was, Does support for an electric baboon-proof fence increase with education/awareness? We hypothesised that an educational video about human–baboon conflict and the success of the Zwaanswyk fence would have a positive impact on support for an electric fence. We tested this by randomising where, in the survey, respondents would be prompted to watch the video. This allowed us to test whether having seen the video affected how respondents responded to the idea of an electric fence. This research method also allowed us to test whether watching the video influenced the amount people would be willing to pay as a levy for the fence and whether respondents considered the fence to be supportive of the welfare of baboons.

### 2.2. Study Area

The survey was conducted in the coastal suburb of Kommetjie, Cape Town, located at 34°08′ S, 18°19′ E on the Cape Peninsula, Western Cape Province, South Africa (Figure 2). There are approximately 3000 people living in Kommetjie, most of whom would be described as well-educated, upper-middle-class citizens of European descent. Kommetjie lies below the Slangkop Mountain, which forms part of the home range for the Slangkop troop of baboons (*n* = 41 individuals), which falls mainly within the Table Mountain National Park. The dominant vegetation type in the Park is fynbos. Baboons are able to obtain all their nutritional requirements from natural vegetation [19] but often prefer the calorie- and protein-rich alien vegetation and human food [17,31]. For this reason, the Slangkop troop frequently enters residential suburbs and commercial/light industrial areas that border the Park and provide access to these anthropogenic food sources. Currently, the City of Cape Town funds a program designed to keep baboons out of all urban areas. The program employs approximately 70 field rangers who work from sunrise to sunset every day of the year using a combination of their physical presence and shouts/whistles to herd troops away from the urban edge [22]. Only if troops persist in attempting to enter urban areas are the rangers then permitted to use paintball guns to enhance their deterrent ability.

Figure 3 provides a visualisation of where an electric fence could be located to keep baboons out of Kommetjie. The proposed fence would be built on the existing firebreak, which serves to prevent fires in the natural vegetation of Slangkop Mountain from encroaching into the suburb. This image was included in the educational video contained within the survey questions; the complete version of the video is available in the Appendix A. The video, using evidence from GPS movements of baboons before and after the installation of the Zwaanswyk fence [26], emphasised that baboons would be able to forage right up to the fence rather than be chased away from land close to settlements as is currently the case.

### 2.3. Survey Sampling and Structure

The survey was emailed to all Kommetjie residents on the Kommetjie Residents’ and Ratepayers’ Association emailing list (*n* = 1192). This represented about a third of all Kommetjie residents (population size: 3341) (City of Cape Town, 2011 census data). The survey was open between 26 November and 20 December 2020. Both versions of the survey should have taken respondents no more than ten minutes to complete, which included watching the video and answering the questions.

At the beginning of the survey, respondents were required to give consent to participate and were informed that doing so was voluntary and that they would be guaranteed anonymity. Research Ethics clearance was granted by the Faculty of Science Research Ethics Committee at the University of Cape Town prior to data collection (approval code: FSREC 065—2020).

Respondents had to answer questions in the order they were presented, and it was not possible to go back and change an answer. Two versions of the survey were created. Both versions of the survey opened with two attitudinal questions relating to the extent of problems experienced with baboons and level of happiness with the presence of baboons in Kommetjie. In version 1 (henceforth known as ‘video_end’), these questions were followed by most of the other questions, with the video placed towards the end of the survey. In version 2 (henceforth known as ‘video_start’), the video was presented immediately after the first two questions, and then the rest of the questions were asked after the video. Respondents were randomly allocated a version of the questionnaire. Those assigned video_start became the experimental group, and those that were assigned video_end became the control group.

The advantage of this approach is that by randomising whether respondents received the ‘treatment’ (video towards the start) or the ‘control’ (video towards the end), we were then able to estimate the effect of having seen the video on how key questions were answered. See [33] for a discussion of the principles of randomised controlled trial design.

Both survey versions (video_end and video_start) had 14 identical questions, but respondents who were randomly assigned to video_end had an additional repeat question asking them again about their attitude towards the baboon-proof electric fence. This allowed us to see if watching the video might have contributed to people changing their minds about the electric fence. Analyses were performed in Stata (SE 15.1) and Excel, and graphs were produced in R [34]. A 5% significance level was used in these analyses.

### 2.4. Data Analysis

Following published principles for comparing different response scales [35], we created dummy variables from the Likert scale answers. The first, indicating support for the fence, allocated a score of 1 to respondents who reported that they ‘strongly supported’ or ‘supported’ the idea of an electric fence to address the baboon issue. Those who reported that they did not support the fence (whether this was strongly held or not) or were ‘neutral’ were allocated a score of 0. A second dummy variable, indicating opposition to the fence, was created that allocated a score of 1 to those who did not support the fence and a 0 to those who supported it or were neutral.

A probit regression model was used to test the main hypothesis that respondents who watched the educational video would have a higher average marginal probability of supporting the electric fence and a lower average marginal probability of not supporting it. The key independent variable—whether respondents had watched the video before answering the survey questions—was coded 1 for those that had watched it towards the beginning of the survey (video_start) and 0 for those that watched it towards the end (video_end).

We hypothesised that attitudes and perceptions towards baboons and baboon-proof electric fences would likely influence the probability of respondents supporting the fence or not. A further regression model was included to control for this. Respondents were asked, “How do you feel about the presence of the baboon troop in the village?”, and responses were captured on a 5-point Likert scale where 1 = very happy, 2 = happy, 3 = neutral, 4 = unhappy, and 5 = very unhappy. The prediction was that the happier people were about having baboons in the village (officially a suburb), the less support there would be for the fence, as a fence essentially inserts a barrier between humans and baboons.

Age and gender were investigated to see if these could predict the level of support for a fence. We did not expect there to be any continuous relationship between age and support for the fence, but we hypothesized that people over 60 (general retirement age in South Africa) would spend more time at home and therefore be more exposed to baboons regularly and hence that they would likely be more supportive of a baboon-proof electric fence. A dummy variable was created in which those over 60 years old and of retirement age were coded as 1 and those under 60 years old as 0.

Although gender has generally been a poor predictor of attitudes [36], a study looking at public perceptions towards “pest management” of squirrels in the United Kingdom found that males were the most accepting of controls to reduce invasive grey squirrels from causing damage to other valued species [37]. We thus also controlled for gender. Those identifying as men were coded as 1, and those identifying as female or non-binary were coded as 0. This was based on the rationale that in terms of how these groups are treated in society, non-binary individuals are more likely to have outcomes more similar to those of females than males, as they do not benefit from a perceived traditionally masculine gender identity [38]. Average marginal effects were calculated to measure the change in the dependent variable as a function of a change in a certain independent/explanatory variable while keeping any other covariates constant.

We also tested two further propositions: that respondents who watched the educational video at the start would have a higher average marginal probability of agreeing that the electric fence would improve the welfare and conservation status of baboons, and that such respondents would have a higher average marginal probability of supporting a monthly levy towards the construction of a fence.

Those who were randomly assigned the video_end version of the questionnaire were allocated additional questions that allowed them to consider, once again, whether they support (or not) the electric fence. Fisher’s exact test was used to test for significant differences between the pre- and post-video support for the electric fence for these respondents. This method follows that of a recent study [39], where a short video-based educational intervention was used to explore whether it could influence clinical trial participation in adolescents and young teens. Participants in the study answered a pre-test survey, viewed a 10 min video, and then completed a post-test survey to assess changes in attitude and intention to participate in future trials. Results of this study showed that intention to participate in a clinical trial was increased by an absolute 18%.

We had no prior expectation as to whether gender would influence whether respondents changed their minds. One study found that gender failed to predict support for baboon deterrent techniques in Cape Town [26], whilst another found that males were more accepting of invasive squirrel controls in the United Kingdom [37]. Neither study directly addressed how education can change attitudes. Our study allowed some initial exploration of the role gender might play in affecting ‘conversion’ following an educational intervention. A dummy dependent variable was generated called ‘Changed mind’ where those respondents who originally did not support the fence or were neutral (‘strongly do not support’, ‘do not support’ and ‘neutral’) subsequently changed their mind to support the fence (‘strongly support’ or ‘support’) were coded as 1. Those that supported the fence initially and continued supporting the fence and those that did not support the fence or were neutral initially and remained that way after watching the video were coded as 0. Probit regression was used to investigate whether gender might predict conversion.

Our study also included a qualitative dimension. Thematic analysis was used to analyse the open-ended responses to the survey question “Please comment on why you do not support a baboon-proof electric fence in Kommetjie if it is built and maintained by conservation authorities and there are no additional costs to you”. The aim was to explore the reasons why people may be against the fence if money is not a limiting factor. A systematic approach was used to find patterned responses or themes within the dataset based (with adaptations) on a protocol provided by Braun and Clarke [40]. This involved familiarisation with the data, reading all the responses to the question to gain a good understanding of the information provided, identifying general themes, and creating response categories based on each of these themes. The responses were recorded on a Microsoft Excel spreadsheet.

## 3. Results

The survey had a response rate of 15.1%. A total of 181 respondents over the age of 18 agreed to participate, out of which there were 166 complete responses and 15 partial responses. The distribution of respondents per survey version was approximately equal (video_end: *n* = 89, video_start: *n* = 92). The breakdown of respondents for each gender and age range for video_end and video_start is shown in Table 1.

### 3.1. Quantitative Results

There was a higher percentage of support for the fence amongst respondents that watched the video at the start (video_start) than those that watched the video at the end (video_end) (Figure 4). For video_start respondents that confirmed they had watched the video (*n* = 77), nearly half of all respondents (49%) strongly support or support the fence. Watching the video before answering the question about support for the fence significantly (*p* = 0.043) raised the average marginal probability of supporting the fence by 15% (Table 2). Watching the video first also significantly (*p* = 0.021) reduced the average marginal probability of not supporting the fence by 17%.

Respondents who expressed a preference for not having baboons in the suburb were significantly more likely to support the fence (*p* = 0.003) and less likely to not support it (*p =* 0.014). Even when controlling for how people feel about the presence of baboons in the suburb, the impact of the educational video remained strong, with the average marginal probability of supporting the fence increasing by 16% (*p* = 0.024) and reducing the marginal probability of not supporting the fence by 18% (*p* = 0.013) (Table 2). Neither age nor gender had statistically significant coefficients, and the models with these additional controls were not reported in Table 2.

Video_end respondents showed a statistically significant increase in support (from 36% to 50%) for the fence (Fisher’s exact = 0.000, 1-sided Fisher’s exact = 0.000) when given an opportunity to watch the video and answer the same questions (Figure 5). All those who supported the fence the first time supported it the second time, and nine people who had not supported it the first time shifted to supporting it after watching the video.

Table 3 shows that responses at the extreme ends of the scale (‘strongly do not support’ and ‘strongly support’) remained similar pre- and post-watching the video, while there was a substantial change in the responses within the ‘do not support’ and ‘support’ categories, with 55.6% and 62.5% of respondents increasing their relative level of support, respectively, after watching the video. People who were neutral before watching the video (*n* = 6) were split in their opinions after watching the video, with 50% (*n* = 3) increasing their support and 50% (*n* = 3) reducing their level of support.

The probit multiple regression showed that being female significantly (dy/dx = 0.189, SE = 0.081, *p* = 0.02) raised the average marginal probability of someone changing their mind in favour of supporting the fence by 19%. Of those who converted to supporting the fence, 88.8% (8/9) identified as female and only one identified as male.

The happier people were with baboons being in the suburb, the less likely they were to support the fence and vice versa. This trend was the same for both video_end and video_start respondents. Only 19.4% (6/31) of respondents who had watched the video at the end and had said they were happy with baboons in the Kommetjie suburb (31/90) supported the fence. For those that said they were unhappy with baboons in the suburb (45/90), 46.7% (21/45) supported the fence. Our results also show that across all levels of being happy (happy, neutral, not happy) with baboon presence in the suburb, the baseline level of support for the fence was higher for respondents who watched the video at the start than it was for those who watched the video at the end. For the respondents that were neutral about baboon presence in the suburb, support for the fence varied from 28.6% for those who watched the video at the end compared to 57.14% if they watched the video at the start.

In total, 41% (74/181) of all respondents said that they did not support a baboon-proof fence even if it was paid for by authorities; 59.5% (44/74) of these respondents had watched the video at the end, and the remaining 40.5% (30/74) had watched the video at the start. Respondents were then given the opportunity to answer an open-ended question stating a reason for their objection(s). Of the total sample (*n* = 181), 70 respondents (38.7%) entered a response to the question: 34 (48.6%) identified as female, 30 (42.9%) identified as male, and 6 (8.6%) people identified as non-binary.

Figure 6 shows that for video_end respondents (*n* = 88), 34% strongly agree or agree that a baboon-proof electric fence improves the welfare and conservation status of baboons, 34% strongly disagree or disagree that a fence improves the welfare and conservation status and 32% are neutral. For those with the video_start version of the survey and who had confirmed they had watched the video (*n* = 76), 58% strongly agree or agree that a fence improves the welfare and conservation status of baboons, 25% strongly disagree or disagree that it improves the welfare and conservation status, and 17% are neutral.

Watching the video significantly increased (*p* = 0.001) the average marginal probability of agreeing that the fence improves the welfare and conservation status by 23%. Watching the video first reduced the average marginal probability of not agreeing that the fence improves the welfare and conservation status by 9%, but this was not statistically significant (*p* = 0.198) (Table 4).

There was a very small increase in support (2%) for paying a monthly levy to construct and maintain a baboon-proof fence amongst respondents that watched the video at the start compared to those that watched the video at the end (Figure 7). For those with the video_end version of the survey (*n* = 88), 21% strongly support or support paying a monthly levy to construct and maintain a fence, 67% strongly do not support or do not support paying the levy and 13% are neutral. For video_start respondents who confirmed that they had watched the video (*n* = 75), 23% strongly support or support paying a monthly levy, 53% strongly do not support or do not support paying a levy and 24% are neutral.

Watching the video increased the average marginal probability of supporting the payment of a levy by 2%; however, this was not statistically significant (*p* = 0.732). Watching the video first reduced the average marginal probability of not supporting the payment of a levy by 14%, but this was also not statistically significant (*p* = 0.066) (Table 5).

### 3.2. Qualitative Results

After conducting a thematic analysis of the open-ended responses, four main themes/dimensions were identified within the final sample relating to objections toward the fence even if it is paid for by conservation authorities: “Pro-wildlife”, “Pro-human”, “Effectiveness of fence” and “General concerns”. Within the main themes, 14 sub-themes were identified and are discussed below (Table 6 and graphical representation in Figure 8).

Many respondents reported that an electric fence would be aesthetically displeasing, i.e., they believed it would be unsightly and take away from the natural beauty of the area. One respondent referring to the fence said, “*It would be a hideous blight on the landscape*”, and another that, “*It will be a terrible eyesore*”. Considering the 17 respondents that mentioned this objection, 41.2% (*n* = 7) had watched the video before answering the question and had seen an example of what the fence in Kommetjie would possibly look like. A further objection is that residents are concerned that they would feel fenced in and that the atmosphere of the suburb would be ruined. One respondent said, “*… the idea of fencing in the village is outrageous. Then you should go and live in a gated community*”, and another that, “*The village atmosphere is being ruined by endless interventions by “authorities” who have no regard for the residents*”.

Slangkop Mountain is a popular hiking spot for Kommetjie residents; therefore, another concern was that a fence would restrict access to the mountain for leisure purposes. The educational video mentioned that there would be gates allowing access to the mountain, and 80% (4/5) of those that expressed concern were respondents who had only watched the video at the end. A number of respondents were concerned about costs associated with the fence, expressing views that the authorities would do better spending the money elsewhere. One respondent said, “*Authorities could use that money to improve many other things e.g., maintaining roads in Kommetjie*”, and another that it will “*cost massive amounts of money that the country simply does not have at the moment*”.

The second most frequently occurring theme (12.9%) was that wildlife should be able to roam freely and that baboons had ranged in Kommetjie and the surrounding areas long before humans settled there. One respondent said, “*We have to keep in mind that the baboons where (sic) here in Kommetjie LONG before humans arrived. So I think it’s highly unfair towards them*. And another, that “*Baboons came first, humans should discipline themselves and adapt to give baboons space they need. Arrogance to impose restrictions on nature*”.

Related to this theme is the view that Kommetjie residents should take responsibility for baboon-proofing their homes and gardens so that baboons are discouraged from entering the suburb, thereby putting the onus on residents rather than using a fence to keep the baboons out of the whole area. Below is a typical comment relating to this theme.


*“It is unnecessary—one cannot manage a wild animal. Instead, one should strive to manage the behaviour of the humans in the system (as a crucial part of the social-ecological system). Baboons are manageable when the residents ensure that houses and waste is baboon proof. This strategy has worked for the past 35 years that I have lived here and will continue to work if residents are educated and comply to some very basic ‘rules’”.*


Many respondents said they were concerned about how the fence might affect other wildlife and were worried that it would fragment the landscape, potentially blocking important ecological corridors. One respondent said that “*This kind of intervention leads to increased fragmentation of habitat, which is not only a problem for the large furry species like baboons! The ethos of Kommetjie is to act as a corridor for the remaining wildlife; to fence in these animals would do irreparable damage*”. Many respondents voiced concerns about how the fence would impact other species in the area, such as caracals, porcupines and mongooses. Many of these concerns were from respondents (11/15—73.3%) who watched the video at the end and had not been given a chance to learn that porcupine tunnels could be built into the fence design.

Some respondents regard the fence as being cruel and inhumane, expressing concerns about baboons and other wildlife facing possible electrocution and other injuries. All these comments were from respondents who watched the video at the end and did not learn that no other animals have ever been injured along the Zwaanswyk fence. The preference for authorities to seek alternative solutions before building a fence in Kommetjie was raised by six respondents, suggesting alternative measures such as planting food and providing water points on Slangkop Mountain or relocating the baboons to other areas.

Respondents (13.7%) stated that baboons are intelligent animals and will be able to breach or circumnavigate the fence, thus rendering it ineffective as a management strategy. One respondent said, “*These animals are so intelligent that they will still find a way around the fence, so a BIG NO to fencing us all in and out*”, and another that, “*The fence as proposed is open on each end and I don’t believe it will keep baboons out of Kommetjie*”. The educational video showed the positioning of the fence, indicating that it would have open edges—which is different from the Zwaanswyk fence that has a single-entry point where the main road enters the suburb. This sub-theme was slightly more popular with respondents who watched the video at the beginning (10/19, 52.6%).

Other concerns are that the fence will be ineffective due to power outages (such as ‘load-shedding’) or because the topography of Slangkop Mountain is not conducive to success. One respondent said, “*I’m concerned that due to the difficult rocky terrain building and especially maintaining an effective fence will be almost impossible. There is a strong likelihood that it will be an expensive disaster*”.

The final sub-theme within this dimension relates to concerns regarding the maintenance of the fence and the possibility that it will be vandalised. A few respondents worried that parts of the fence might be stolen, and others that maintenance of the fence would be an onerous task. One respondent, who had watched the video at the start, went so far as to say: “*You are a monkey if you think any fence in the country can be maintained, it’s a joke, there are fences everywhere, not one has been maintained, look at the reserve, look at Imhoff, look at the navy, look at Soetwater. Show me one fence that has done its job for more than six months…*”.

## 4. Discussion

This study provides evidence that an educational video about the potential for an electric fence to reduce negative interactions between humans and baboons can increase support for an electric fence. The study has implications for the management of baboons in the area, but the study is of more general interest, too, in that it provides rare empirical evidence of the efficacy of an educational intervention. It is widely acknowledged in conservation that engaging with local stakeholders is essential for many biodiversity conservation projects, yet relatively little attention has been paid to how stakeholder engagement impacts conservation outcomes, in part because this can be difficult to evaluate [41]. To the best of our knowledge, there are no other studies that empirically test whether education can improve stakeholder support for a particular management measure. This is despite previous studies acknowledging that evidence-based decision-making is critical for conservation actions, especially for conservation conflicts where there are implications to public safety and wildlife populations [29].

Watching the video first increased the average marginal probability of supporting an electric fence by 15% and reduced the average marginal probability of not supporting the fence by 17%. Neutral responses were similar between the control and treatment groups (12% and 14%, respectively), suggesting that the video intervention was influential enough to change attitudes from not supporting to supporting the fence rather than simply pushing respondents towards a neutral stance.

Within the broader context of conservation conflicts, the results are significant, as managing a species is considerably more effective if there is collaboration, agreement and support for management measures between key stakeholders such as local community members, policy and decision-makers, scientists and management practitioners [37]. Conversely, disagreements between key stakeholders can act as a barrier and cause delays in the implementation of effective management measures [26].

The pre- and post-test analysis of respondents who received the video_end version of the questionnaire showed that support for the fence increased significantly because of seeing the video. The pre- and post-test conversion results thus provide greater confidence in the robustness of the results gained from the randomised controlled trial method. Watching the video first increased the average marginal probability of agreeing that a fence improves the welfare and conservation status of baboons by 23% and the average marginal probability of support for the fence by 15%. This indicates that there are potentially other factors underlying the reasons behind stakeholders not supporting the fence despite acknowledging that the fence will improve baboon welfare and conservation status.

One of those factors, and a major concern that is often raised regarding the use of fences to mitigate human–baboon conflict, is cost. Fences, although extremely effective at keeping baboons out of urban areas, can be expensive to build and maintain [22,28]. The Zwaanswyk model, where residents pay a monthly levy towards a baboon-proof electric fence (which covered the initial building cost and now covers all associated maintenance costs), prompted us to explore whether watching the video would influence the attitudes of Kommetjie residents towards the payment of a monthly levy. The results show that the video was not able to predict support for the levy payment and that watching the video increased the average probability of residents supporting this payment by only 2% (and this was not statistically significant). This may reflect shortcomings in the video or that we were unable to control for income or other financial considerations of respondents. It is important to note that almost 40% of the respondents were at or close to retirement age (>60 years) and thus would have limited scope for increasing their income. This financial constraint may have influenced their willingness to support a levy for the construction and ongoing maintenance of a baboon-proof fence, but in the absence of data on their financial status, this was not a variable we could explore directly.

In a study looking at the human–elephant conflict in Nepal, residents’ willingness to pay towards conflict mitigation programs was found to be positively related to income and education [42]. Although the survey did have a question relating to income range, over a third of residents chose the option ‘prefer not to say’, which rendered it useless in the analysis. A more robust model is needed to examine the factors involved in willingness to pay towards a fence, where the use of an educational video may help to garner support for a levy amongst those that can afford to pay but still have non-monetary concerns/misconceptions regarding the fence. This is bolstered by the fact that watching the video first reduced the average marginal probability of not supporting the payment of a levy by 14 percentage points.

Although the lack of support for a levy payment might be related to missing information pertaining to socioeconomic status, it was concerning, from a management point of view, that 41% of respondents reported that they would not support an electric fence even if it were built and maintained by conservation authorities and there would be no additional costs to them. This challenged the assumption that residents paying for an electric fence was a major barrier. The qualitative analysis of the responses revealed that objections to the fence were not unidimensional but covered a broad spectrum ranging from pro-welfare to pro-human oriented concerns whilst also revealing many practical issues residents had regarding the effectiveness of the fence.

This suggests that promoting support for a baboon-proof electric fence will require a nuanced and tactical approach when engaging with stakeholders. This is emphasised by the fact that the top four recurring sub-themes from the thematic analysis were that baboons might be able to circumnavigate the fence, that wildlife should be able to roam freely, the fence will be aesthetically displeasing and the concern for other wildlife. It is evident that addressing a single dimension will not be adequate from a managerial perspective. The results do show, however, that a higher percentage of those who do not support the fence, even if there is no cost to them, are respondents who had not watched the video before answering the question (60% compared to 40% that had watched the video). This provides some insight into the potential shortcomings and limitations of the video as an educational tool to promote support for a fence whilst also highlighting some of the more successful aspects.

The sub-theme with the highest frequency of occurrence is the acknowledgement by respondents that baboons are intelligent animals that may potentially breach the fence, either by finding novel ways to pass over it or by transgressing the fence at its edges. A potential limitation of the video was that the virtual placement of the fence showed that it would be open at either end. What the video failed to convey explicitly was that field monitors would remain employed and that they could ensure that baboons cannot breach the fence, either via access points to the mountain or at the fence edges. The fact that this concern comes almost equally from those that had watched the video and those that had not indicates that the video probably failed to educate on this issue and hence allay residents’ concerns in this regard.

Pro-human-related concerns, where respondents feel that the fence will impede their enjoyment of life, are more difficult to address regardless of the educational video. There is not much scope to make an electric fence less of an eyesore. The technical specifications required for the fence to be effective against baboon incursions (electrified strands, Bonnox^®^ mesh, etc.), as detailed by O’Riain and Hoffman [28] and Kaplan [26], will always contrast somewhat with the natural surroundings of the area. Stakeholders who hold steadfast views that the fence is unattractive or that it will ruin the feel of the Kommetjie suburb are less likely to change their attitude towards a fence, even when presented with the relevant facts about how it will improve the welfare and conservation status of baboons.

To increase support for the fence for those with pro-human concerns, it may ultimately be more impactful to use an educational tool with an emphasis on what escalating baboon conflict is likely to do to disrupt their lifestyle, i.e., they will have to ensure that homes are fully baboon-proofed, they cannot easily enjoy outdoor dining, etc. Research into the effects of information on attitudes toward suburban deer management found that tailoring information towards stakeholder concerns centred on either the “effectiveness” or “humaneness” of contraception was more likely to influence attitudes towards the use of contraception as a management measure [43]. A similar approach may be more effective at aligning those stakeholders with a pro-human mentality towards ultimately supporting a fence.

Pro-wildlife stakeholders who enjoy having the baboons in the suburb predictably did not support a baboon-proof electric fence (the most effective management measure for reducing human–baboon conflict).

## 5. Conclusions

By utilising two different experimental methods, a randomised controlled trial and a pre- and post-test design, we were able to show that the education of residents on an intervention that is advocated by experts improves their probability of stakeholder support. Watching an educational video, which intentionally addresses common misconceptions about baboon-proof fences, increased the probability of supporting the fence by 15% and reduced the probability of not supporting the fence by 17%. In the broader context of negative human–wildlife interactions, the findings of this study support the popular refrain in conservation and wildlife management literature that education is a powerful tool for improving the acceptance of proposed conservation actions that are known to benefit both wildlife and communities [29].

The successes and failures of the video in this study were revealed in open-ended responses to questions about support for the fence, even if it is paid for by conservation authorities. These lessons can be used to improve future educational videos and further motivate research that seeks to understand the values and ideologies of stakeholders, which underpin support or lack thereof for management interventions. In practical terms, such research would provide conservation practitioners with actionable information that would allow for more effective communication and satisfaction among community members.

Through the production of a short video, a wide audience can be educated on the benefits and costs of the intervention allowing them to make a more informed decision on whether they would support its use. It is, however, unclear from this brief study as to the durability of the results and how well the intention to support the fence will translate into future real-world support for the fence. Current research on education in conservation stresses the need for ongoing education on key issues and not simply once-off interventions such as that attempted in this study. Further studies investigating the persistence of attitude change because of an educational intervention are necessary. We have, however, shown that at least in the short-term, Kommetjie residents were more aligned with supporting a baboon-proof electric fence to minimise negative human–baboon interactions following the viewing of an educational video, and thus being nudged towards a positive conservation solution.

## Figures and Tables

**Figure 1 animals-13-02125-f001:**
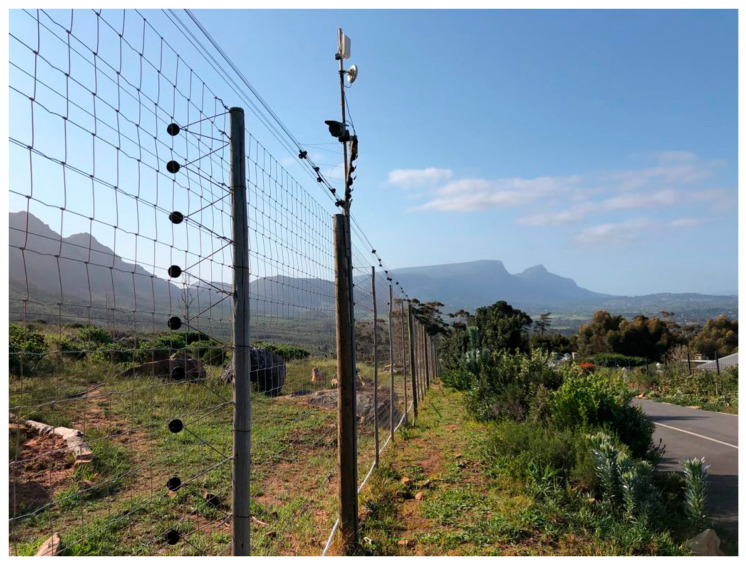
The baboon-proof electric fence in the suburb of Zwaanswyk in Cape Town. Photo © Jessica Burnette, 2020.

**Figure 2 animals-13-02125-f002:**
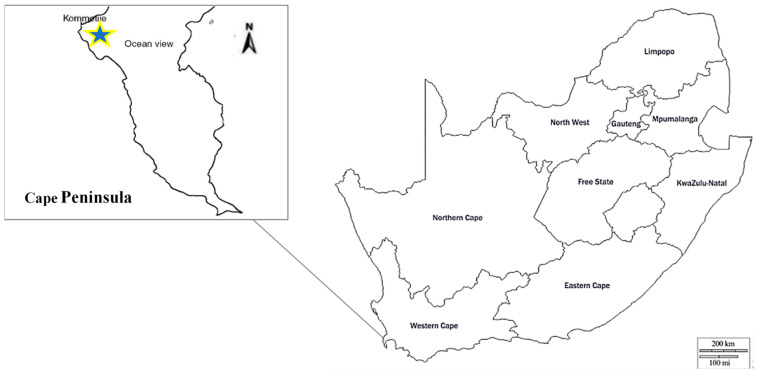
Map showing the location of the study site, Kommetjie (star), on the Cape Peninsula in South Africa. Map image modified from © d-maps.com (accessed on 2 March 2021) [32].

**Figure 3 animals-13-02125-f003:**
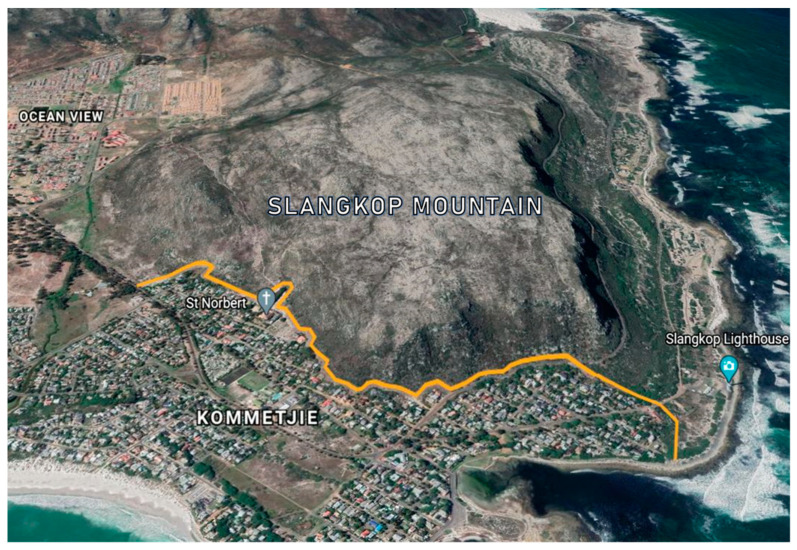
Google Earth image visualising where a baboon-proof electric fence in Kommetjie could be situated. Maps data: Google, © 2020 Maxar Technologies AfriGIS (Pty) Ltd.

**Figure 4 animals-13-02125-f004:**
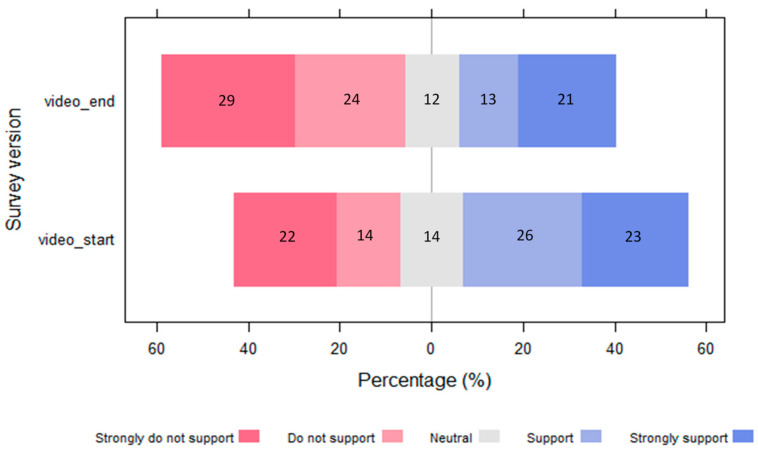
Diverging bar plot showing a positive difference in the level of support for the fence for respondents who had watched the video first (video_start) compared to those who did not see the video prior (video_end) to answering the survey question “do you support a baboon-proof fence?”. The percentages for each Likert response are indicated within the stacks.

**Figure 5 animals-13-02125-f005:**
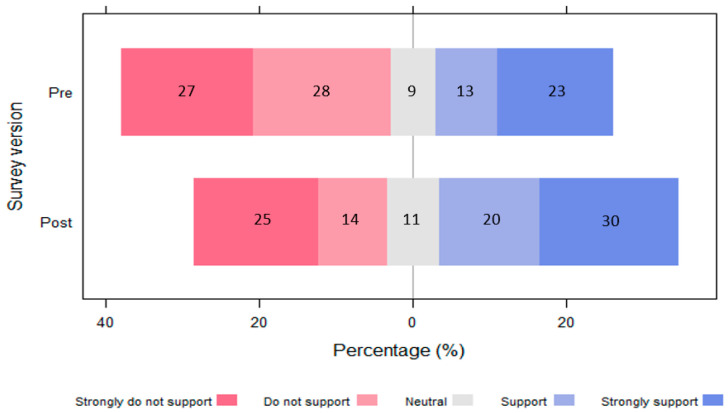
Diverging bar plot showing a positive difference in the level of support for the fence for respondents that watched the video at the end (video_end) and were given another chance to state their level of support after watching the video (*n* = 64). Those who indicated they had not watched the video were excluded.

**Figure 6 animals-13-02125-f006:**
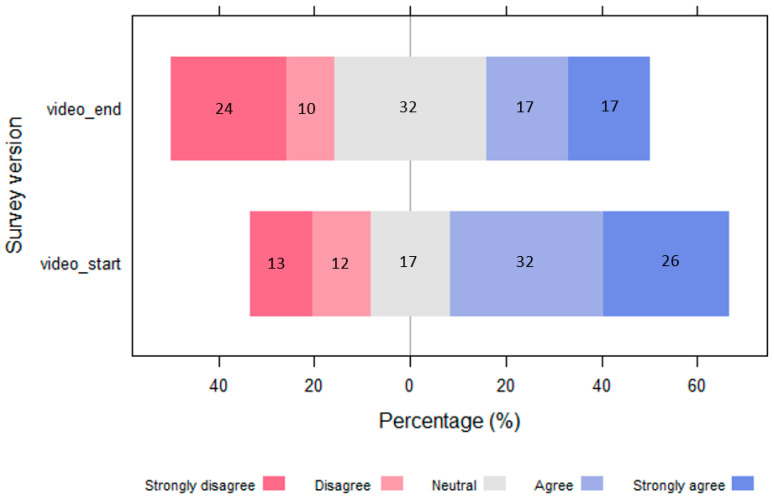
Diverging barplot showing a positive difference between video_end and video_start respondents in response to the question “do you agree or disagree that a baboon-proof electric fence improves the welfare and conservation status of baboons?”. The percentages for each Likert response are indicated within the stacks.

**Figure 7 animals-13-02125-f007:**
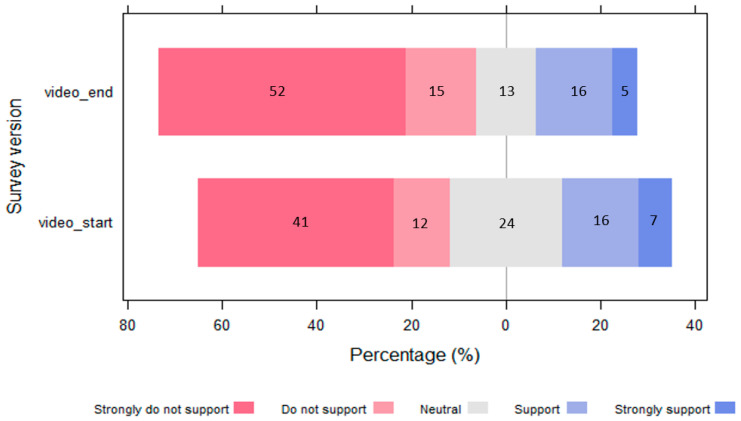
Diverging barplot showing a difference in the level of support for those that watched the video at the end (video_end) and those that watched it at the start (video_start) in response to the question “Please indicate your willingness to pay a monthly levy to construct and maintain an electric fence”. The percentages for each Likert response are indicated within the stacks.

**Figure 8 animals-13-02125-f008:**
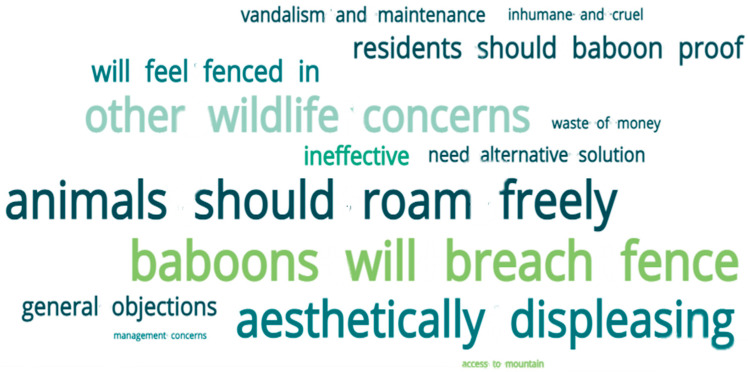
Phrase cloud for the sub-themes relating to objections to the fence based on respondents saying that they do not support a fence if it is paid for by conservation authorities. The size of the phrase represents the frequency of occurrence from both video_end and video_start respondents.

**Table 1 animals-13-02125-t001:** Demographic characteristics of respondents for gender and age and the distribution of these characteristics by survey version (video_end and video_start).

Characteristic of Respondent		Response Category	Video_End	Video_Start	Total	%
Gender		Female	36	44	80	44.2
		Male	43	37	80	44.2
		Non-Binary	4	2	6	3.3
		No answer	9	6	15	8.3
Age		18–20	1	0	1	0.6
		21–29	1	0	1	0.6
	Non-retirement age	30–39	4	6	10	5.5
		40–49	22	9	31	17.1
		50–59	23	29	52	28.7
		60–69	19	22	41	22.7
	Retirement age	70 and above	14	16	30	16.6
		No answer	9	15	15	8.3

**Table 2 animals-13-02125-t002:** Results of probit models showing that watching the educational video first increased the average marginal probability of supporting the fence and reduced the average marginal probability of not supporting the fence (reporting average marginal effects). Partial responses were included for respondents who had answered this question before bailing out of the survey. Those with the video_start survey version who reported that they had not watched the video were excluded from the analysis.

	Dependent Variable: 1 = Support the Fence, 0 = Other		Dependent Variable: 1 = Do Not Support the Fence, 0 = Other	
	Video (dy/dx)	Video + Happiness (dy/dx)	Video (dy/dx)	Video + Happiness (dy/dx)
1 = Watched the video first	0.147 * (0.073) (*p* = 0.043)	0.160 * (0.071) (*p* = 0.024)	−0.167 * (0.072) (*p* = 0.021)	−0.178 * (0.072) (*p* = 0.013)
1 = Not happy with baboon presence		0.206 * (0.069) (*p* = 0.003)		−0.175 * (0.071) (*p* = 0.014)
Prob > chi2	0.051	0.005 *	0.028 *	0.006 *
Number of observations	167	167	167	167
Akaike’s information criterion (AIC)	226.64	220.78	229.31	225.83

Standard errors are in parentheses. * *p* < 0.050.

**Table 3 animals-13-02125-t003:** Changes in response to the level of support for the fence pre- and post-watching the video. The number and percentage of responses that stayed the same, increased or decreased within each Likert response category are stated.

Original Likert Score: Support for Fence	Total Number of Responses	Level of Support	Change in Response	Percentage Change (%)
1—Strongly do not support	17	Same	12	70.6
		Increase	5	29.4
2—Do not support	18	Same	5	27.8
		Increase	10	55.6
		Decrease	3	16.7
3—Neutral	6	Same	0	0.0
		Increase	3	50.0
		Decrease	3	50.0
4—Support	8	Same	3	37.5
		Increase	5	62.5
		Decrease	0	0.0
5—Strongly support	15	Same	12	80.0
		Decrease	3	20.0

**Table 4 animals-13-02125-t004:** Probit model testing whether watching the video first increased the average marginal probability of agreeing that a fence improved welfare and conservation status of baboons and reduced the average marginal probability of not agreeing (reporting average marginal effects: dy/dx). Partial responses were included for respondents who had answered this question before bailing out of the survey. Those with the video_start survey version who reported that they had not watched the video were excluded from the analysis.

	Dependent Variable: 1 = Agrees Fence Improves Welfare, 0 = Other	Dependent Variable: 1= Does Not Agree Fence Improves Welfare, 0 = Other
	(dy/dx)	(dy/dx)
1 = watched the video first	0.230 ** (0.069) (*p* = 0.001)	−0.091 (0.071) (*p* = 0.198)
Prob > chi2	0.002 *	0.203
Number of observations	164	164

Standard errors are in parentheses. ** *p* < 0.001, * *p* < 0.050.

**Table 5 animals-13-02125-t005:** Probit model testing whether watching the video first increased the average marginal probability of supporting a monthly levy to construct and maintain an electric fence and reduced the average marginal probability of not supporting a levy (reporting average marginal effects).

	Dependent Variable: 1 = Supports a Monthly Levy, 0 = Other	Dependent Variable: 1 = Does Not Support a Monthly Levy, 0 = Other
	(dy/dx)	(dy/dx)
1 = watched the video first	0.022 (0.064) (*p* = 0.732)	−0.135 (0.074) (*p* = 0.066)
Prob > chi2	0.732	0.074
Number of observations	163	163

**Table 6 animals-13-02125-t006:** Sub-themes relating to reasons why residents do not support a baboon-proof electric fence in Kommetjie if it is built and maintained by conservation authorities and there are no additional costs to them for video_end and video_start respondents.

Main Themes/ Dimensions	Sub-Themes Relating to Objections to the Fence	Frequency Video_End	Frequency Video_Start	Total Frequency	Total (%)
Pro-human	Aesthetically displeasing	10	7	17	12.2
	Residents do not want to be fenced in/ruin feel of suburb	5	4	9	6.5
	Concern about access to mountain/trails	4	1	5	3.6
	Cost/waste of money	2	4	6	4.3
Subtotal:					26.6
Pro-wildlife	Baboons were here first/wildlife should roam freely	12	6	18	12.9
	Concern for other wildlife/fragmentation of habitat	11	4	15	10.8
	Residents to take responsibility (i.e., baboon-proofing)	6	3	9	6.5
	Inhumane/cruel	6	0	6	4.3
	Alternative solution preferred, i.e., relocation	2	5	7	5.0
Subtotal:					39.5
Effectiveness of fence	Baboons intelligent/will breach or circumnavigate fence	9	10	19	13.7
	Fence ineffective, i.e., load-shedding ^1^, topography	6	2	8	5.8
	Fence will be vandalised/Maintenance concerns	2	5	7	5.0
Subtotal:					24.5
General concerns	General objection, i.e., just do not want fence	4	4	8	5.8
	No confidence in management authorities	3	2	5	3.6
Subtotal:					9.4
				139	100

^1^ Load-shedding is the switching off of parts of South Africa’s electric grid in a planned and controlled manner due to insufficient capacity or to avoid a countrywide blackout.

## Data Availability

Data requests can be directed to the corresponding author.

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
