# Peer review of "On the Fence: The Impact of Education on Support for Electric Fencing to Prevent Conflict between Humans and Baboons in Kommetjie, South Africa"

_animals, 2023, doi:10.3390/ani13132125_

Round 1
Reviewer 1 Report
Please see attachment

Reviewer 2 Report
Please see my comments in the attached document.

Reviewer 3 Report
The authors present a relevant manuscript about the use of electric fences to control human-baboon conflict and the perceptions regarding this topic. I believe conservation measures like this and people's perceptions should be carefully evaluated and investigated to establish general guidelines to be followed in different geographical locations. Therefore, the authors should receive credit for their work. I do not have any suggestions or corrections to do to this work. I believe it is explained and interpreted and detailed. Here you have my suggestions:
Introduction
The sentence where the authors present the aims of their study should be in a separate paragraph.
Methods: The methodology should be divided in subsections to provide an easier understanding (e.g Location; statistical analysis....)
1. What is the main question addressed by the research?
This research addresses people's reactions and the impact of people's education regarding the use of electric fences specifically to minimize the human-baboon conflict in this critical area.
2. Do you consider the topic original or relevant in the field? Does it address a specific gap in the field?
As far as I know, the topic of the human-baboon conflict in Cape Town is not new at all. There have been many tries and strategies to find the best solution to this problem, considering several factors, such as animal welfare, people's safety, the health of animals and humans, people's perception etc... Nevertheless, it remains necessary to figure out and evaluate the current strategies to avoid human-animal conflict (as electric fences) regarding all the mentioned factors. Considering the aims of the special issue "Living with Non-human Primates: Conflicts, Perceptions and Coexistence" to which this manuscript has been submitted, as well as how well it is written, I believe it reunites what is needed to be considered for publication, with minor details to be corrected.
3. What does it add to the subject area compared with other published material?
I believe I have answered this question in my precious answer. The human-wildlife conflict is growing as fast as the human population in city centres is. It is necessary to publish material on every perspective of a strategy (as an electric fence), including human perception.
4. What specific improvements should the authors consider regarding the methodology? What further controls should be considered?
As I mentioned in my report, it is usual to have subsections in the methodology section and it helps the readers to understand how the research was conducted. That is my suggestion to the authors for improvement.
5. Are the conclusions consistent with the evidence and arguments presented and do they address the main question posed?
Yes, I believe so. Besides interpreting the study and comparing it with others, the authors provided critical thinking about the study's limitations and what can be done in the future regarding the subject (for instance L 628-635). This is what should be done in the DIscussion and COnclusions of every research. That is why I have nothing else to add regarding these sections.
6. Are the references appropriate?
Yes. I believe citations should be in numerical form but I also think authors can change this before publication.
7. Please include any additional comments on the tables and figures.
I sincerely have nothing to criticise regarding tables and figures. They are well structured, and clearly illustrate the location, the type of fences, and the statistical results of the study. I also think it is a positive thing to have the video as supplementary material.
